# Integrative Approaches in the Management of Hypertrophic Cardiomyopathy: A Comprehensive Review of Current Therapeutic Modalities

**DOI:** 10.3390/biomedicines13051256

**Published:** 2025-05-21

**Authors:** Marco Maria Dicorato, Gaetano Citarelli, Francesco Mangini, Rossella Alemanni, Miriam Albanese, Sebastiano Cicco, Cosimo Angelo Greco, Cinzia Forleo, Paolo Basile, Maria Cristina Carella, Marco Matteo Ciccone, Andrea Igoren Guaricci, Ilaria Dentamaro

**Affiliations:** 1Cardiology Unit, Interdisciplinary Department of Medicine, Polyclinic University Hospital, University of Bari “Aldo Moro”, 70121 Bari, Italy; cinzia.forleo@uniba.it (C.F.); paolo.basile@uniba.it (P.B.); m.carella31@phd.uniba.it (M.C.C.); marcomatteo.ciccone@uniba.it (M.M.C.); andreaigoren.guaricci@uniba.it (A.I.G.); ilaria.dentamaro@gmail.com (I.D.); 2Cardiology Division, San Paolo Hospital, 70132 Bari, Italy; gaetanocita@libero.it; 3Cardiology Division, Miulli Hospital, 70021 Acquaviva delle Fonti, Italy; francescomangini.78@libero.it; 4Cardiac Surgery Division, Casa Sollievo Della Sofferenza Hospital, 71013 San Giovanni Rotondo, Italy; rossellaalemanni@gmail.com; 5Division of Cardiology, V. Fazzi Hospital, 73100 Lecce, Italy; albanesemiriam91@gmail.com; 6Internal Medicine Unit “Guido Baccelli”-Arterial Hypertension Unit “Anna Maria Pirrelli”, Department of Precision and Regenerative Medicine and Jonic Area (DiMePReJ), Polyclinic University Hospital, University of Bari “Aldo Moro”, 70124 Bari, Italy; 7Cardiology Division, Veris Delli Ponti Hospital, 73020 Scorrano, Italy; cosimoangelo.greco@gmail.com

**Keywords:** hypertrophic cardiomyopathy, left ventricular outflow tract obstruction, medical therapy, cardiac myosin modulators, mavacamten, gene therapy, genome editing, myectomy, septal ablation, atrial fibrillation, sudden cardiac death prevention

## Abstract

Hypertrophic cardiomyopathy (HCM) is often associated with left ventricular outflow tract (LVOT) obstruction, which affects a substantial proportion of patients. This obstruction results from a range of anatomical abnormalities involving both the valvular and subvalvular structures. Pharmacological therapies play a pivotal role in the management of LVOT obstruction, with a range of drug classes exhibiting distinct mechanisms of action. Beta-blockers, including atenolol and nadolol, are considered the first-line treatment due to their ability to reduce heart rate and myocardial contractility and enhance diastolic filling. Non-dihydropyridine calcium channel blockers, such as verapamil and diltiazem, are utilized as second-line agents when beta-blockers are ineffective or contraindicated. Disopyramid, a Class 1A antiarrhythmic agent, is employed for patients who do not respond to initial therapeutic interventions and can reduce LVOT gradients. Recent advancements in cardiac myosin modulators, such as Mavacamten and Aficamten, offer targeted therapies by modulating myosin–actin interactions to reduce LVOT gradients and improve symptoms, with promising results from clinical trials. Although gene therapy is still in its nascent stages, it has the potential to address the genetic basis of HCM by employing techniques such as genome editing, gene replacement, and the modulation of signaling pathways. For patients exhibiting severe symptoms or demonstrating unresponsiveness to medical treatment, invasive therapies, such as septal reduction therapy and alcohol septal ablation, are considered. Ultimately, the treatment and prevention of atrial fibrillation and sudden cardiac death are two key points of HCM management in both obstructive and non-obstructive forms. This review aims to provide an overview of current pharmacological and invasive strategies, as well as emerging therapies, in the management of HCM.

## 1. Introduction

Hypertrophic cardiomyopathy (HCM) is a genetic heart disease characterized by marked wall thickness that is unexplained by overload conditions. Despite its historical prevalence, the efficacy of standard pharmacological therapy remains uncertain due to the paucity of randomized clinical trials demonstrating its effectiveness [1]. The natural history of HCM is marked by the development of heart failure (HF), arrhythmias, and sudden cardiac death (SCD), exhibiting clinical presentation heterogeneity ranging from asymptomatic conditions to severe manifestations. The initial step in the therapeutic approach is the identification of left ventricular outflow tract (LVOT) obstruction. In obstructive forms, defined by an LVOT gradient >30 mmHg, symptoms manifest as exertional dyspnea/fatigue, chest pain (angina), syncope/presyncope, and HF. In these cases, the therapeutic objective is to reduce LVOT obstruction. In non-obstructive forms, treatment objectives include the reduction of HF and of atrial fibrillation (AF) occurrence, as well as the prevention of cardioembolic events [2,3]. Diagnostic progress has led to a precise distinction between obstructive and non-obstructive forms and between HCM and phenocopies [4,5,6,7,8,9,10,11]. Actually, there is a wide range of conditions that simulate HCM, with disparate mechanisms. For example, cardiac amyloidosis is a condition in which misfolded proteins infiltrate the extracellular space of the cardiac muscle. The thickening of myocardial walls is a consequence of the accumulation of glycosphingolipids in individuals diagnosed with Fabry disease. The differential diagnosis between these conditions is imperative for the implementation of effective pharmacological therapy [12,13]. In this regard, the standard drugs that have been used for years to treat this condition include beta-blockers, calcium channel blockers, and dysopiramid. ATPase inhibitors are novel agents that have exhibited the capacity to modify the natural progression of HCM, exerting their effects on the heightened contractility of the myocardium [1]. They have marked a pivotal shift in the therapeutic approach for HCM. Gene therapy constitutes a novel strategy that aims to precisely modify specific DNA sequences through various techniques, such as genome editing, gene replacement, allelic silencing, and the modulation of signaling pathways. These treatments offer significant potential, although further research is required to fully realize their clinical applicability [14]. Invasive strategies for treating LVOT obstruction include septal reduction therapy, which encompasses ventricular septal myectomy and septal ablation [1]. Another salient point pertains to the management of non-obstructive forms and AF, which constitutes the most prevalent arrhythmia in this demographic. In addition to conventional antiarrhythmic medications, catheter ablation has been shown to be both safe and effective, although large, randomized trials are lacking [15]. In the end, the prevention of SCD in this category of patients is mandatory and can be achieved only through an implantable cardioverter–defibrillator (ICD). The objective of this review is to analyze the different therapeutic approaches available in the context of HCM, encompassing those that have been largely demonstrated to be effective to the emerging ones. It is imperative to comprehend such a multifaceted approach, particularly in a pathology such as HCM, which exhibits a wide range of phenotypes and grades of severity and necessitates comprehensive treatment strategies.

## 2. Pharmacological Therapy in Obstructive Forms

LVOT obstruction is present in one-third of patients [16] and is caused by many factors and by the anomalous anatomy of valvular and subvalvular apparatuses. Describing the pathophysiological mechanisms of LVOT obstruction is beyond the scope of this review. The drugs for the management of LVOT obstruction are various, with different characteristics [1,3].

### 2.1. Beta-Blockers and Non-Dihydropiridine Calcium Channel Blockers

Beta-blockers (BBs) represent the most prevalent class of drugs employed in the treatment of HCM, with their utilization being predicated on the attenuation of sympathetic activity, culminating in a reduction in heart rate and ventricular contractility. This, in turn, engenders an enhancement of diastolic filling, a diminution of oxygen demand, and a reduction in the exercise-induced LVOT gradient [17]. The first beta-blocker utilized was propanolol, as evidenced by a small non-randomized study demonstrating a reduction in the gradient and an improvement in symptoms [16]. Its efficacy in the treatment of angina and the control of heart rate has led to its frequent use in pediatric patients diagnosed with obstructive HCM (OHCM) [18]. However, it should be noted that propanolol is not the drug of choice for adult patients. Atenolol, with doses up to 150 mg, is the recommended treatment for adults with obstructive HCM and hypertension. Similar to propanolol, it is effective in managing angina and effort dyspnea, and it is more potent in regulating heart rate. Nadolol, similar to atenolol, is effective in managing LVOT obstruction and reducing non-sustained ventricular tachycardia (NSVT) episodes [19]. Bisoprolol and metoprolol are the drugs of choice in cases of HCM with heart failure and in end-stage HCM. However, Dybro et al. have shown that metoprolol reduced LVOT obstruction at rest and during exercise, provided symptom relief, and improved quality of life also in patients with OHCM [20]. It is imperative to note that the dosage of these medications must be gradually increased until they are tolerated by the patient. The primary constraints associated with the utilization of BBs pertain to their adverse effects, which include bradycardia, hypotension, and airway reactions [21]. Non-dihydropiridine calcium channel blockers (ND-CCBs) are utilized when beta-blockers are contraindicated or not tolerated. These agents represent a second-line treatment option due to their reduced efficacy compared to beta-blockers. The rationale for using this class of drugs is, once again, the negative inotropic effect, obtained with a different mechanism than BBs: the reduction in calcium (Ca^++^) intracellular inflow by blocking calcium channel [22]. In animal models, calcium channel blockers have demonstrated efficacy in treating alterations associated with the R403Q mutation in the MHY6 gene, even in the absence of hypertrophy [23]. The negative chronotropic effect of calcium channel blockers leads to improved diastolic filling, reduced oxygen demand, and improved symptoms. However, the vasodilation caused by these drugs can lead to dangerous effects, including increased dynamic intraventricular gradient and peripheral edema [3,22]. Numerous studies have shown the possibility of severe adverse effects with verapamil, which limits its use in patients with severe obstruction or increased ventricular filling pressure [24]. Despite these adverse effects, a recent study showed that in patients with HCM, either obstructive or non-obstructive, verapamil was not associated with a higher incidence of cardiovascular events than BB therapy, suggesting that it may be a safe alternative to BBs in these patients [25,26].

### 2.2. Disopyramid

The utilization of disopyramid is recommended by European Guidelines (Class I) for patients who continue to experience symptoms or demonstrate non-responsiveness to beta-blockers or calcium channel blockers [3]. Disopyramid is classified as a Class 1A antiarrhythmic agent, exhibiting a negative inotropic effect. This cardiac effect of disopyramid has been demonstrated to improve symptoms and reduce the LVOT gradient [27,28]. Disopyramid is administered in conjunction with beta-blockers or calcium channel blockers, as it can facilitate AV conduction and increase the risk of high-rate atrial fibrillation (AF) episodes. It is imperative to inform patients of the necessity to avoid the use of concomitant medications associated with QTc prolongation [29]. The limitations of disopyramid include its gradual loss of efficacy over time [30,31], the absence of a long-acting formulation, necessitating frequent dosing (every 6–8 h), which can cause patient discomfort, the potential for QTc prolongation requiring ECG monitoring, anticholinergic effects (e.g., xerostomia), and, in male patients, prostatism and urinary tract symptoms [28]. In instances of disopyramide intolerance, cibenzoline, a class Ia antiarrhythmic agent, has been employed, demonstrating efficacy in reducing LVOT obstruction and enhancing diastolic function [32]. In a recent real-world study, Maurizi et al. [33] demonstrated that the best responders to disopyramid therapy were younger, had a smaller left atrium, a less severe LVOT gradient, and a higher LV ejection fraction, with a safe arrhythmic profile. These findings are in line with the drug physiopathological effect, mainly driven by its negative inotropic effect, which is the main mechanistic driver of reduction in LVOT gradients. In Figure 1 and Table 1, the mechanism of action and properties of the principal drugs used in the management of OHCM are represented.

### 2.3. Cardiac Myosin Modulators: Mavacamten and Aficamten

Mavacamten and Aficamten are two new drugs used to treat LVOTO in OHCM. These treatments represent a turning point in the scenario of cardiomyopathies management, being able to change the natural history of the pathology [34]. HCM is characterized by an exaggerated interaction between myosin and actin filaments in the heart muscle. The mechanism of action of Mavacamten works by reducing ATP hydrolysis, which decreases the number of myosin heads available to interact with actin filaments. It stabilizes myosin in a super-relaxed state, preventing cross-bridge formation. This effect can also downregulate the metabolic pathway linked to hypertrophy and fibrosis in HCM. Early animal testing of Mavacamten led to successful Phase 1 and 2 clinical trials. The PIONEER-HCM trial [35], a Phase 2 study, demonstrated that Mavacamten reduced the LVOT gradient and improved symptoms, NYHA class, oxygen demand, and pro-BNP levels. The EXPLORER-HCM trial [36], a Phase 3 study, involved 251 patients with HCM and LVOT gradients > 50 mmHg. Mavacamten treatment led to a reduction in LVOT gradient and improved diastolic function, left atrial volume, and LV mass index. However, 6% of patients had reduced left ventricular function, which improved after stopping treatment. About one-third of patients did not respond to Mavacamten, suggesting varying responses among patients, unrelated to genotype. The VALOR-HCM trial [37], which focused on patients eligible for septal reduction therapy (SRT), showed that only 18% of Mavacamten-treated patients maintained eligibility for SRT after 16 weeks, indicating improvement in symptoms. Results from the ongoing MAVA-LTE trial [38] suggest that Mavacamten remains effective in the medium term. A recent study by Wheeler et al. demonstrated that Mavacamten was effective in improving functional capacity, LVOTO, symptom burden, and biomarkers, independent of the use of beta-blockers [39]. Subsequent meta-analyses have confirmed the safety and efficacy of this drug in OHCM [40,41]. Mavacamten has also been recently tested on symptomatic HCM patients without LVOTO. The medication was found to be well tolerated by the majority of subjects, resulting in a significant decrease in biomarkers (NT-proBNP and troponin levels) [42]. These findings provide a foundation for additional research targeting the non-obstructive HCM population. The second cardiac myosin inhibitor, Aficamten, has a shorter half-life (3–4 days) compared to Mavacamten (6–8 days) and reaches the steady state more quickly (2 weeks vs. 6–8 weeks for Mavacamten). Aficamten does not interact with CYP450 enzymes, different from Mavacamten, which affects CYP2B6 and CYP3A4. The REDWOOD-HCM trial [43] demonstrated that Aficamten effectively reduced LVOT gradients in patients with NYHA class II or III symptoms, with no significant adverse effects, except for one patient who discontinued treatment due to reduced left ventricular function. REDWOOD-HCM cohorts 1 and 2 showed that Aficamten may reduce LVOT gradient safely in patients already in treatment with BBs and/or ND-CCBs. On the other hand, in REDWOOD-HCM cohort 3, Aficamten demonstrated to be an effective add-on treatment in patients who do not respond to all other medical therapies [44]. Aficamten has been demonstrated to be effective in improving peak oxygen uptake, symptoms, exercise capacity, and NT-pro-BNP levels [45,46]. A substudy of the SEQUOIA-HCM trial has also demonstrated the efficacy of Aficamten in improving cardiac remodeling, particularly in regard to left ventricular mass, wall thickness, and left atrial size [47]. The ongoing phase 3 MAPLE-HCM trial is evaluating Aficamten as both a first-line therapy for newly diagnosed OHCM and as an alternative to standard drugs (BBs and CCBs) [48]. Both cardiac myosin ATPase inhibitors have shown promise in clinical trials, with continued research needed to optimize their use. The mechanism of action of ATPase inhibitors is illustrated in Figure 2.

## 3. Gene Therapy

HCM is an autosomal dominant disorder that is genetically heterogeneous. It results from over 1500 identified mutations across at least 15 distinct genes [49]. Since advancements in DNA sequencing technology have been made, genetic testing and counseling are now strongly recommended [3,50]. Given the genetic basis of HCM, novel therapeutic opportunities have emerged, including gene therapy. This approach entails the precise modification of specific DNA sequences. Gene therapy is predicated on several methodologies. One of the first tested techniques was genome editing, which involves the precise modification of specific DNA nucleotides using cluster regularly interspaced short palindromic repeats (CRISPR)-Cas9 endonucleases. These endonucleases can cleave DNA at specific sites, allowing for the precise alteration of genetic material [51]. The first application of genome editing was in a study involving human embryos. In this study, human sperm carrying a pathological MYBPC3 mutation were injected into oocytes with a normal MYBPC3 allele. The study demonstrated the remarkable capacity of CRISPR-Cas9 to produce a normal MYBPC3 allele in 72% of the embryos [52]. Subsequent studies employing genome editing in animal models yielded encouraging results, underscoring its potential for therapeutic applications [53,54]. Another technique is gene replacement, which is used when the mutation causes deficiency or complete absence of the protein, as in the case of haploinsufficiency. This technique is particularly suitable for MYBPC3, the most common mutation in HCM. The objective of this approach is to introduce a functional gene copy that can synthesize a functional protein. This approach does not directly address the mutation; rather, it provides an exact copy of the gene, thereby reducing the production of the abnormal protein [23]. Two strategies are employed to achieve this objective. The first strategy involves the introduction of a wild-type MYBPC3 DNA copy (cDNA) by viral vectors, resulting in an increase in mRNA and wild-type myosin-binding protein C levels in animal models, with this increase being dose dependent [55]. The second strategy utilizes RNA transcription, wherein the pre-RNA is spliced together with gene therapy products, which consist of sequences of wild-type exons, to create a repaired mRNA. Allele-Specific Silencing is another possible approach in this setting. It consists of modifying the mRNA, emulating the mRNA alteration system that is inherently present in cells. It involves exon skipping, which entails the introduction of oligonucleotides that are complementary to specific sequences of the pre-RNA into cells by viral vectors. These oligonucleotides attach themselves to these sequences, impeding the action of the splicing regulatory protein and preventing the sequences from being inserted into the mature RNA. This results in the formation of an RNA devoid of the mutation [56]. This technique finds application when an allele of a gene carries a mutation and the other allele is normal [51]. RNA interference is particularly well suited for conditions driven by gain-of-function mutations, such as MYH7, but its applicability to the entire spectrum of HCM, including MYBPC3-related cases, is not yet clear [1]. The modulation of signaling pathways is another technique that involves interference with the pathways that regulate the contraction and relaxation of the cardiac muscle. Two therapeutic targets have been identified for HCM: sarcoplasmic/endoplasmic reticulum Ca^2+^ ATPase 2a (SERCA2) and myosin regulatory light chain (myosin RLC). In HCM, a reduction in SERCA2 levels has been observed, suggesting that the introduction of this mutation by viral vectors may prove beneficial [14]. The mutation of myosin RLC has been linked to an aggressive form of HCM. In animal models, the introduction of the aforementioned viral vectors has been demonstrated to result in a reduction in left ventricular hypertrophy and an enhancement in cardiac function [57,58,59]. Figure 3 presents the novel gene therapy approaches previously described.

Notwithstanding the encouraging results, there are numerous limitations and challenges that must be considered. The utilization of viral vectors, which are innocuous, has the potential to stimulate the immune system. However, many patients may possess neutralizing antibodies that could render the vectors ineffective. The CRISPR/Cas9 system, in particular, poses a series of challenges due to its potential for off-target activity and the induction of unintended mutations. This potential for off-target effects is a subject of intensive research, with the development of selective RNA guides aimed at mitigating these risks. Despite the encouraging results, the translation of these findings into clinical practice remains in its nascent stages, with the safety and ethical considerations being paramount [14].

## 4. Invasive Strategies: Surgical Myectomy and Septal Ablation

LVOTO is defined as a peak LVOT gradient of ≥30 mmHg. However, the threshold for invasive treatment is generally considered to be ≥50 mmHg (resting or provoked by exercise or Valsalva) in patients with severe symptoms (NYHA functional class III-IV) despite maximally tolerated pharmacological treatment [3]. Invasive therapy may be considered also for patients with mild symptoms (NYHA class II) who do not respond to medical therapy. This therapy is recommended in expert centers with demonstrable low procedural complication rates [60]. Invasive treatment for LVOTO involves septal reduction therapy (SRT) when the anterior septal thickness is equal to or greater than 15 mm. The two main techniques for SRT are ventricular septal myectomy and alcohol septal ablation (ASA). An assessment checklist is crucial for the evaluation of candidates for invasive treatment. It is imperative to assess alternative causes that can explain symptoms (e.g., obesity, respiratory disease, coronary artery disease, anemia, thyroid disease, arrhythmia, amyloid, and right ventricular outflow tract obstruction). Imaging plays a pivotal role in evaluating the underlying mechanism of obstruction (e.g., SAM-related, sub-aortic membrane, anomalous papillary muscle insertion, and accessory mitral valve tissue), the function of the mitral valve (MV), and the distribution and severity of hypertrophy [3]. Surgical myectomy has long been regarded as the gold standard for treating symptomatic patients with significant LVOTO. It effectively reduces symptoms, maintains long-term survival rates, and can improve left ventricular function by alleviating the obstruction caused by muscle hypertrophy [61,62]. Notably, a systematic review indicated that the five-year survival rates post-myectomy are high, reportedly around 98.9% [61]. In contrast, alcohol septal ablation, a less invasive technique, has also gained recognition for its safety and efficacy. Studies indicate that ASA can provide similar symptomatic relief and improve the LVOT gradient comparable to myectomy, yielding notable long-term benefits and survival rates that align with those of surgical myectomy [62,63]. Comparative analyses suggest that both procedures offer comparable outcomes regarding mortality and functional status post-operation. A systematic review concluded that recent evidence shows no significant difference in cumulative outcomes, including survival and stroke incidence, between the two interventions [61,62]. Furthermore, ASA has emerged as a prudent option for older patients or those with comorbidities that may render them unfit for surgical intervention, highlighting its importance in a diverse patient population struggling with HCM [63,64]. Recent case reports and analyses have demonstrated the nuanced decision making involved in choosing between these two approaches. For instance, ASA is often considered for patients who are not ideal candidates for myectomy due to anatomical variations or significant comorbidities [65,66]. Conversely, in cases where LVOT obstruction is concomitant with other cardiac diseases necessitating surgical intervention—such as anomalous papillary muscle, elongated anterior mitral leaflet, or coronary artery disease—surgical intervention is the optimal approach. This is due to the fact that it facilitates the correction of both functional and anatomical abnormalities in a single procedure [67]. Additionally, procedural advancements, such as the use of high-precision imaging and mapping techniques, have enhanced the efficacy and safety profiles of ASA, aligning well with contemporary surgical goals of minimally invasive practices [68]. Furthermore, ongoing investigations aim to explore combined approaches or novel pharmacological adjuncts that may aid in the management of HCM and influence decisions regarding surgical interventions [69]. From these studies, it emerged that ASA was associated with a higher rate of reoperation and less reduction in the LVOT gradient [70], with a higher risk of atrio-ventricular (AV) block, requiring permanent pacemaker (PM) implantation. The risk of AV block following surgery and ASA is higher in patients with pre-existing conduction disease, although recent data suggest that the long-term outcome of patients after ASA with an implanted PM is similar to those without a PM [71,72]. Conclusively, both surgical myectomy and alcohol septal ablation represent critical elements of the therapeutic landscape for hypertrophic cardiomyopathy. A clear understanding of the distinctions, advancements, and practical applications of these interventions enables healthcare providers to develop tailored, patient-centric management strategies.

### 4.1. Ventricular Septal Myectomy

This procedure consists of creating a rectangular trough in the basal septum below the aortic valve and then extending it distally to beyond the point of the mitral leaflet–septal contact. The classic Morrow myectomy was described for the first time in 1975 [73]. In the classic Morrow procedure, LVOTO is relieved by resecting relatively small sections of muscle tissue in the proximal ventricular septum, which widens the LVOT and partially decreases the hydrodynamic drag forces along with a “Venturi” effect, resulting in SAM (Figure 4). Many variations of this procedure have been reported with varied efficacies [74,75]. With regard to the modified Morrow procedure, the incision was extended from the conventional approach involving a midventricular excision, progressing leftward toward the mitral valve annulus and downward to the bases of the papillary muscles (Figure 4c). In all regions where the papillary muscles were fused to the septum or ventricular free wall, separation was performed, resulting in the severing or removal of abnormal chordal structures, muscle clusters, and fibrous connections of the mitral leaflets to the ventricular septum or free wall. This myectomy, considered comprehensive, results in the expansion of the LVOT area and the redirection of forward blood flow. Consequently, the drag and Venturi effects on the mitral valve are eliminated [76]. Pre-operative determinants of a good long-term outcome are age < 50 years, left atrial size < 46 mm, absence of AF, and male sex [3]. Typically, septal myectomy is approached via the transaortic route. Transaortic septal myectomy is a well-tried procedure, with wide and long follow-ups; in addition, the aorta incision is not arrhythmogenic [74,75,77]. On the other hand, transaortic septal myectomy is characterized by a limited operating field, with limited visualization of the MV [67]. This makes this procedure prohibitive in case of small aortic annulus (i.e., infants and young children); in such instances, the modified Konno procedure has been reported to provide equally satisfactory long-term results for basal obstruction [78,79,80]. Therefore, other surgical approaches are being tested as transapical and transmitral exposure. The transapical approach is a safe and effective approach for relief of midventricular obstruction [81,82]. The technique can be combined with transaortic myectomy for patients with left ventricular outflow obstruction at both levels [82]. The transmitral approach is feasible and reliable for the treatment of certain types of OHCM cases. Potential advantages of transmitral myectomy include a panoramic view of the septum and mitral subvalvular apparatus and the ability to simultaneously address MV pathology [83]. The latter is preferable in patients with diffuse hypertrophy, abnormal papillary muscle orientation, and MV abnormality. The transmitral approach allows for aggressive papillary muscle reorientation [84], the enlargement of the LVOT using an autologous pericardial patch for anterior MV leaflet (Figure 4d) [85], and MV repair or replacement. Cardiac imaging provides key information for planning surgery: small aortic root, anterior mitral leaflet length > 30 mm, intraventricular septum thickness < 19 mm or neutral septum, and mitro-aortic angle < 120° leaning to the transmitral approach. The management of MV is a big challenge to HCM surgery. MV replacement is more common in non-specialized HCM centers than in specialized HCM centers. Valve replacement eliminates SAM and associated mitral regurgitation, as well as the outflow tract gradient, but the addition of MV replacement with or without myectomy increases the hospital mortality rate (>10-fold) and length of hospitalization compared with patients undergoing isolated septal myectomy. Further, when intervention on the valve at the time of myectomy is needed because of intrinsic mitral disease, every effort should be made to repair the valve because early and long-term mortality is worse in patients with prosthetic replacement compared with patients who have septal myectomy and MV repair [67]. A very complex category of OHCM patients are those affected by mitral annular calcium (MAC), who represent about 1/10 of the OHCM population: MAC is associated with anterior displacement of mitral coaptation and contributes to the pathophysiology of LVOTO [86]. These patients have worse prognosis and more recurrent MV regurgitation than those without MAC after septal myectomy. Furthermore, a paravalvular leak is often present after replacement [87]. In patients with AF, concomitant ablation using the Cox–Maze procedure can also be performed [3].

### 4.2. Alcohol Septal Ablation and Alternative Methods

ASA involves the percutaneous injection of absolute alcohol into the septal arteries supplying the hypertrophied myocardium, resulting in a controlled myocardial infarction that reduces LVOTO and alleviates symptoms [88,89]. The outcomes of this procedure are similar to surgery in terms of gradient reduction, symptom improvement, and exercise capacity, including younger adults also [3,90]. Due to the variability in the septal blood supply, it is imperative that myocardial contrast echocardiography be performed prior to alcohol injection. Nevertheless, the procedure is not without complications. The administration of substantial quantities of alcohol into multiple septal arteries during catheterization is generally discouraged. This approach is intended to reduce the gradient, but it may result in significant risks, including adverse effects and arrhythmic events [91,92]. The primary risks associated with ASA include the potential for complete heart block, necessitating pacemaker implantation in 10–12% of patients [66]. Notably, careful patient selection and planning, such as considering the distribution of coronary arteries, are crucial for minimizing complications associated with this technique [93]. Variations in alcohol dosage have also been implicated in outcomes, as lower doses may be associated with higher rates of repeat septal reduction therapies due to inadequate myocardial necrosis [62,94]. The decision-making process regarding the use of ASA versus other treatment modalities relies on an interdisciplinary approach, incorporating cardiologists’ insights on the individual patient’s pathology, risks, and expected outcomes [89,92]. A number of alternative approaches have been documented in a limited number of patient groups. These include non-alcohol septal ablation (non-ASA) methods such as microcoils, polyvinyl alcohol foam fragments, and cyanoacrylate, as well as direct intracavitary and intramuscular treatments like radiofrequency and cryotherapy [95,96]. Glue septal ablation (GSA) is a promising approach. Its main advantage is immediate polymerization, which prevents leakage into the left anterior descending coronary artery. This characteristic renders GSA particularly beneficial for patients with collateral circulation to the right coronary artery, for whom alcohol ablation is contraindicated. No significant complications in the clinical experience of this technique have been reported. The procedure leads to an immediate reduction in the LVOT gradient, and this improvement has been demonstrated to be maintained throughout 12 months of follow-up [96]. Furthermore, GSA has been shown to enhance functional capacity according to the NYHA classification and to reduce interventricular septal wall thickness. However, further research is necessary to assess the long-term efficacy and safety of this technique. Also, microcoil embolization has produced promising results, avoiding the toxic effects of alcohol [97]. A lower percentage of complications (e.g., need for PM implantation and ethanol flow to other myocardial regions) has also been reported [98]. Another substance recently employed in septal ablations is polidocanol, a sclerosing agent primarily utilized in the treatment of varicose veins, which features a more uniform distribution within the target vessel. Initial studies indicated a favorable safety and effectiveness profile, with outcomes and complication rates comparable to those of traditional ASA [99]. The utilization of polidocanol holds particular promise in the context of mid-ventricular obstructive HCM, where surgical myectomy and ASA are associated with a heightened risk of complications and technical challenges. Ates et al. demonstrated a significant reduction in the midventricular gradient and interventricular septum thickness, accompanied by an improvement in the NYHA functional class, in 11 patients [100]. These alternative methods have not been directly compared with other septal reduction therapies, and long-term outcome/safety data are not available [96]. In Figure 5, a flow chart for the management of OHCM is represented.

## 5. Management of Non-Obstructive Forms

In patients without obstruction in the LVOT and with a normal ejection fraction, symptoms are secondary to diastolic dysfunction due to LV hypertrophy, stiffness of the wall, a small cavity, and myocardial fibrosis [101]. The initial step in the diagnostic process is the evaluation of the LVEF. If the EF is equal to or greater than 50%, the administration of beta-blockers or calcium channel blockers (e.g., verapamil or diltiazem) is recommended [3]. These medications have been shown to enhance quality of life, alleviate symptoms, extend the diastolic filling time, and mitigate the severity of chest pain [102]. Nitrates represent a second-line therapeutic option, unless concomitant coronary disease is present. When the LVEF is equal to or less than 50%, the therapeutic approach for HF remains consistent, involving the administration of beta-blockers, angiotensin-converting enzyme inhibitors (ACEis), angiotensin receptor blockers (ARBs), mineralocorticoid receptor antagonists (MRAs), sodium-glucose cotransporter 2 inhibitors (SGLT2is), and low-dose diuretics (Figure 6) [3,17,103,104]. Other studies focused on the role of different drugs in non-obstructive HCM with preserved left ventricular ejection fraction. The use of sacubitril/valsartan in symptomatic patients with non-obstructive HCM is currently under study [105]. However, initial findings indicate that a 16-week treatment period with this pharmaceutical agent, while generally well tolerated, does not demonstrate substantial enhancements in exercise capacity or cardiac structure and function. The impact of losartan has been thoroughly investigated in several randomized controlled trials. First reports indicated the efficacy of the drug in halting the progression of hypertrophy and fibrosis [106]. However, this finding was contradicted by subsequent research [107]. Another study demonstrated the beneficial effect of valsartan in improving cardiac remodeling in individuals with sarcomeric HCM [108]. Also, the role of ranolazine in this context remains a subject of debate [109]. Recent trials have not demonstrated any beneficial effects in non-obstructive HCM [110].

Patients with HCM who are completely asymptomatic, devoid of signs of HF, and exhibiting no symptoms due to arrhythmias do not require treatment. However, some patients classified as “asymptomatic” derive benefit from low-dose beta-blockers, particularly during periods of physical exertion. It is imperative to inquire about the presence of symptoms in patients, as they may exhibit self-limiting daily activities. In young patients who engage in regular physical activity, the administration of beta-blockers is recommended to mitigate the potential for an increase in heart rate [111]. Routine ambulatory visits are necessary to monitor the progression of the pathology and to promptly intervene as required [112].

## 6. Management of Atrial Fibrillation

Given the prevalence of atrial fibrillation (AF) as the most commonly observed arrhythmia in patients with HCM, a discussion of its treatment is pertinent. In patients with LVOTO, the presence of AF often results in a more unfavorable prognosis if not addressed promptly [113]. In cases of paroxysmal AF, restoring sinus rhythm is reasonable, given the atrial contribution to ventricular filling. According to established guidelines, amiodarone is recommended as the initial treatment, with subsequent electrical cardioversion. In the long term, sotalol and disopyramid are the first drugs of choice for the prevention of AF recurrences, particularly in patients with LVOTO [114,115,116,117]. It is imperative to avoid other antiarrhythmic class IC drugs, such as flecainide and propafenone, due to their proarrhythmic potential. In cases where the restoration of sinus rhythm is not feasible, it is imperative to implement rate control with beta-blockers (even in cases of a reduced ejection fraction) or calcium channel blockers (only if the ejection fraction is preserved) [114]. The use of digoxin is contraindicated; it may be utilized only in advanced, end-stage HF. According to European Guidelines [3], oral anticoagulation is recommended to reduce the risk of stroke and thromboembolic events in all patients with HCM and AF or atrial flutter (unless contraindicated), independent of the CHADS-Va score [118] (Table 2).

Another important strategy for achieving rhythm control is catheter ablation. Although this invasive technique has demonstrated long-term restoration of sinus rhythm in up to two-thirds of cases, repeat ablation procedures or continued antiarrhythmic therapy are often necessary [3,15]. According to recent data, about 50% of patients are free from AF after 1 year, and about 35% are free from AF after 3 years. These rates are significantly lower compared to the general population [119]. Catheter ablation plays a central role, particularly when pharmacological therapy is ineffective or poorly tolerated. It can also be chosen as the initial strategy according to the patient’s preference [120]. Regarding the ablation strategy, pulmonary vein isolation is less effective compared to the non-HCM population due to the presence of different arrhythmogenic foci. This is primarily due to physiologic features that increase left atrial pressure, leading to modifications in LA structure [113]. Other strategies include creating more extensive lesions to target specific proarrhythmic areas identified via electrogram mapping. Various energy delivery modalities have been tested as well, including radiofrequency and cryoablation, with no significant difference in long-term outcomes [119,121]. Pulsed-field ablation has yielded promising results, although large-scale trials are lacking [122].The overall complication rate of the catheter ablation procedure is about 5%, mainly due to vascular access-related events, such as major bleeding. Severe complications, such as cardiac tamponade and stroke, occur more frequently than in the general population, despite being reported in less than 1% of cases. For this reason, this technique should be performed in high-volume centers by experienced operators [120]. Surgical AF ablation is another possible strategy, especially for patients undergoing open-heart surgery for a myectomy. Although data from a wide range of studies are promising, they are limited by the small number of patients [123].

## 7. Sudden Cardiac Death: Prognostic Factors and Prevention

The annual incidence of sudden cardiac death (SCD) in HCM patients is estimated around 1–2% per year. Heart failure and thrombo-embolism are the main causes, but fatal arrhythmic events also play a significant role: ventricular fibrillation (the most common), asystole, AV block, and pulseless electric activity (PEA) [124]. The application of an Implantable Cardiac Defibrillator (ICD) is the only effective therapy in preventing SCD and is indicated in HCM patients with high SCD risk. Therefore, SCD risk stratification is a fundamental step for the management of such patients. In secondary prevention, ICD implantation is always indicated by both European and American Guidelines [3,67]. Resuscitated cardiac arrest due to ventricular tachycardia or ventricular fibrillation, or previously documented sustained ventricular tachycardia, indicates Class I ICD implantation. Regarding primary prevention, a large number of risk factors have been investigated in several studies [3]: some of them have been included in the “HCM risk score”, an individualized estimated 5-year SCD risk assessment model that incorporates several disease-related risk factors into a logistic regression equation, dividing the population into three groups of risk [125]. The effect of age, the presence of non-sustained ventricular tachycardia (NSVT), the severity of LV wall thickness, the family history of SCD, the history of recent unexplained syncope, the left atrial diameter, and the presence of LVOTO have been evaluated in a large series of studies and finally included in this score for their significant correlation with increased incidence of SCD [126,127,128,129,130,131]. The HCM risk score has a key role in estimating SCD risk, and its use in clinical practice is indicated by European Guidelines [3] in Class I, at the level of evidence B. An ICD in primary prevention is recommended with an indication of class IIa in the presence of a calculated risk of sudden death > 6%/year. With an intermediate 5-year SCD risk (>4%, <6%), ICD can be considered (class IIb), whereas in low-risk patients (HCM risk score < 4%) ICD can be considered (class IIb) only in presence of ≥1 clinical risk factors (CMR-derived LGE > 15% or LVEF < 50%). In the European guidelines, for the first time, directions are suggested for ICD implant in the pediatric population (age < 16 years). The ICD implant is recommended in primary prevention in the presence at least of two known risk factors for sudden death, and it is recommended to consider, as a cut-off of maximum thickness of the left ventricle, a thickness > 30 mm or a z-score > 6 [3]. American Heart Association Guidelines focused on additional risk factors and elaborated an algorithmic approach based on a single-risk factor approach [67,103]. One or more of the recognized risk markers in HCM is considered relevant and major within the overall clinical profile of a clinically diagnosed patient and, thus, sufficient for the consideration of primary prevention ICD implantation. For adult patients with >1 major risk factors for SDC, it is reasonable to offer an ICD. These major risk factors include young SCD in >1 close relatives, massive LV hypertrophy (>30 mm), syncope, LV apical aneurysm with transmural scar or LGE, and LV systolic dysfunction (EF < 50%) [67]. The amount of LGE seems to outperform the HCM risk SCD score and the American algorithm in the identification of HCM patients at increased risk of SCD and reclassifies a relevant proportion of patients [132]. More recently, additional risk markers are being investigated to improve current risk models and aid in clinical decision making. They include T1 mapping and entropy, both obtained via CMR, LV global longitudinal strain, the presence of AF, cardiopulmonary testing, biomarkers and electrocardiography parameters [133,134,135]. Global longitudinal strain (GLS) has been demonstrated to be significantly related to increased SCD risk in HCM patients [133,136]. Also, CMR-feature tracking-derived GLS is a powerful independent predictor of MACE in patients with HCM [137]. As regards biomarkers, N-terminal pro-brain natriuretic peptide (NT-proBNP) is reported to be independently related to an increased risk of SCD [138]. Strong evidence and new perspective studies supplied a large number of SCD risk factors, which can, with a combined and patient-centered approach, estimate the individual risk supporting the choice of the most effective prevention measures. Actually, the ICD indication must always be evaluated considering the psychological and socio-economic aspects and the burden of device-related complications. A CMR-suitable ICD must be favored for these patients to undergo this examination. Less complex devices should be considered in children (single-chamber, single coil, and subcutaneous), taking into account body size and growth. In the case of indication for an ICD, biventricular pacing must be evaluated, above all, in end-stage patients needing pacing. A subcutaneous ICD must be taken into consideration as an alternative to the transvenous defibrillator in patients who do not require anti-bradycardic stimulation, mainly if young or with high infection risk. Wearable defibrillators should be considered in patients who are not currently eligible for an ICD.

## 8. Limitations and Future Perspectives

Recent investigations into multimodal treatment approaches for HCM highlight achievements and limitations, offering insights into future research and clinical practice directions. Although multimodality imaging has advanced significantly, offering detailed insights into disease morphology and functional dynamics and enhancing diagnosis and management strategies, challenges remain in standardizing treatment efficacy across diverse patient profiles. Current studies emphasize the importance of personalized therapies; however, they often lack randomized, large-scale, long-term clinical trials to validate their findings [139]. Furthermore, although multimodal imaging has improved risk stratification and postoperative outcomes [140,141], the complex nature of HCM, particularly its genetic basis, indicates incomplete understanding of the disease pathophysiology, limiting the generalizability of these findings. Myosin inhibitors represent a turning point in HCM management but lack long-term safety data due to limited follow-up [142]. Future research will focus on long-term outcomes of these drugs. The increasing use of genetic testing in cardiomyopathy management is crucial for decision making, with prognostic and therapeutic implications. Various gene therapy approaches are under study, and research will focus on this topic due to the specificity of this treatment approach [143]. Additionally, although surgical options such as septal myectomy and alcohol ablation have demonstrated significant benefits, they are often only applicable to select patient cohorts, thereby limiting their therapeutic impact [144,145]. Future invasive strategies consist of novel techniques aimed at decreasing the incidence of complications. Ultimately, artificial intelligence (AI) and deep learning techniques are becoming increasingly significant due to their ability to improve diagnostic accuracy and personalize treatment. Recent studies have proposed models that use artificial neural networks to integrate genetic data and clinical variables, improving the prediction of phenotypic outcomes and the personalization of treatment plans [146]. Furthermore, applying AI to multimodal imaging is a promising way to assess disease progression and treatment response, facilitating timely interventions [147,148]. As these technologies evolve, they have the potential to transform clinical practices by offering more nuanced evaluations and predictive analytics, which could lead to better outcomes for patients with HCM [149].

## 9. Conclusions

HCM with LVOTO represents a complex clinical challenge, requiring a multifaceted therapeutic approach. Pharmacological treatments, including beta-blockers, calcium channel blockers, and disopyramide, remain fundamental in the management of symptoms and the reduction of intraventricular gradients. The advent of cardiac myosin modulators has emerged as a pivotal development, with the potential to transform the natural progression of the pathology. While gene therapy is still in its nascent stages, its potential to revolutionize the field by addressing the underlying genetic causes of HCM is a promising avenue for future research. For patients exhibiting refractory symptoms, invasive treatments such as surgical myectomy and alcohol septal ablation offer viable alternatives. AF management and SCD prevention represent another critical component of the treatment of these patients, with important implications for outcomes and quality of life. In the contemporary era, the landscape of HCM has undergone significant transformations due to the advent of therapeutic interventions—first of all, Mavacamten—capable of altering the natural progression of the condition and improving its associated symptoms. A personalized and multimodal treatment approach, incorporating both established and emerging therapies (such as genome editing), is pivotal in managing the diverse manifestations of obstructive and non-obstructive HCM. Ongoing research and large randomized clinical trials are imperative to refine these treatments, optimize their use, and explore novel therapies for enhanced patient outcomes.

## Figures and Tables

**Figure 1 biomedicines-13-01256-f001:**
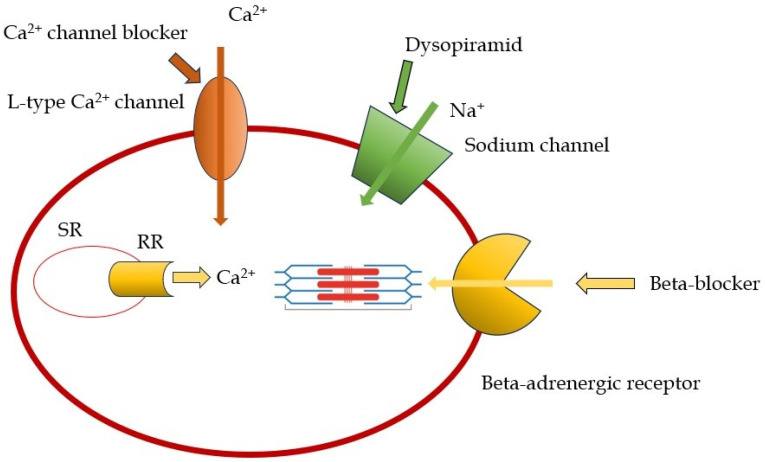
Mechanism of action of the main studied drugs used in the management of hypertrophic cardiomyopathy. SR: sarcoplasmic reticulum; RR: Ryanodine receptor.

**Figure 2 biomedicines-13-01256-f002:**
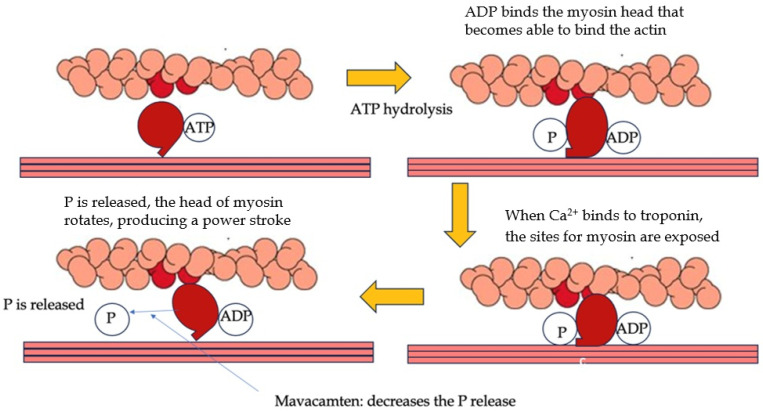
Representation of the mechanism of action of cardiac myosin ATPase inhibitors. ATP: adenosine triphosphate; ADP: Adenosine diphosphate; P: Phosphate.

**Figure 3 biomedicines-13-01256-f003:**
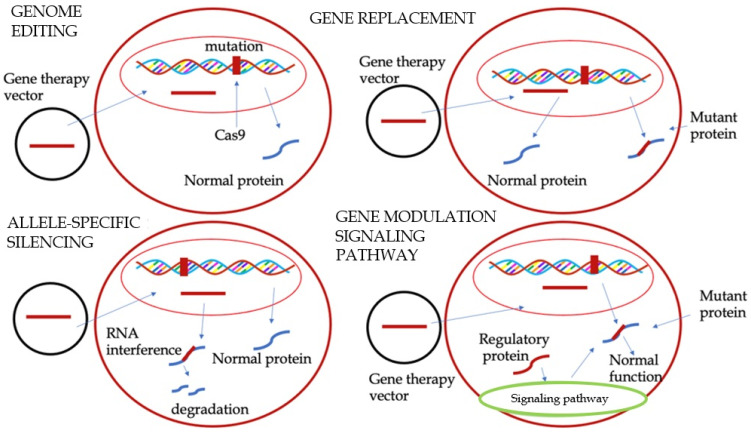
Different gene therapy approaches for hypertrophic cardiomyopathy.

**Figure 4 biomedicines-13-01256-f004:**
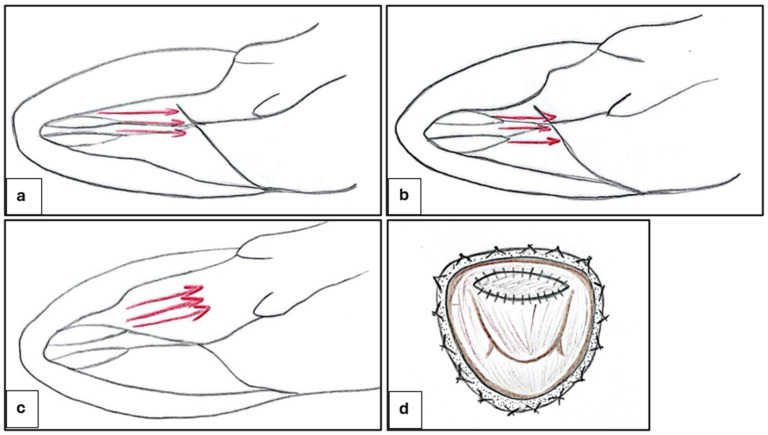
Surgical approach used in hypertrophic cardiomyopathy. (**a**) The hydrodynamic drag forces, in conjunction with the Venturi effect, result in the occurrence of SAM in OHCM. (**b**) The classic Morrow procedure entails the resection of relatively small muscle tissue sections in the proximal ventricular septum, thereby widening the LVOT. However, this procedure does not alter the orientation of resistance forces, resulting in the persistence of SAM. (**c**) The modified Morrow procedure involves extending the classic incision with a midventricular resection, which results in the widening of the LVOT and a concomitant alteration in the orientation of forces, leading to the dissolution of SAM. (**d**) The LVOT is enlarged using an autologous pericardial patch for the anterior MV leaflet. OHCM: obstructive hypertrophic cardiomyopathy; SAM: systolic anterior motion; LVOT: left ventricular outflow tract.

**Figure 5 biomedicines-13-01256-f005:**
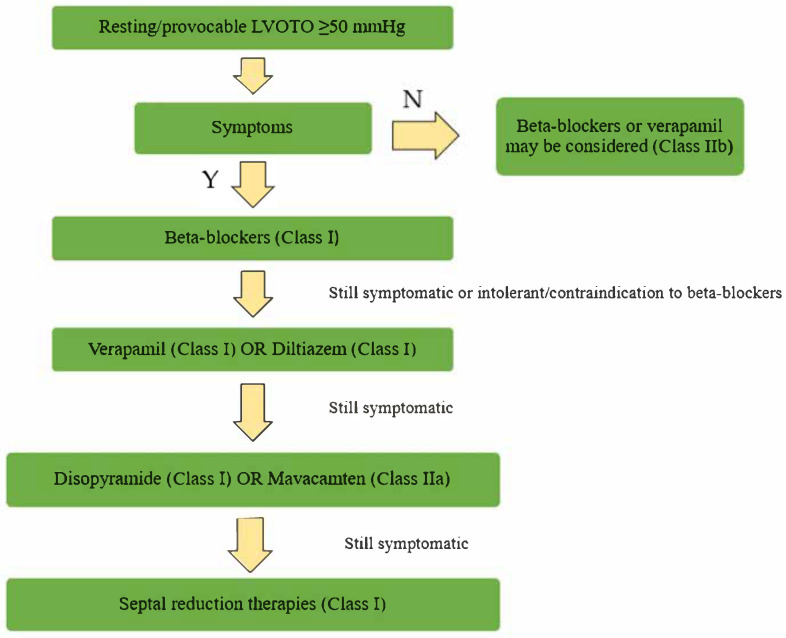
Management of patients with obstructive hypertrophic cardiomyopathy. LVOTO = left ventricular outflow tract obstruction; Y = yes; N = no.

**Figure 6 biomedicines-13-01256-f006:**
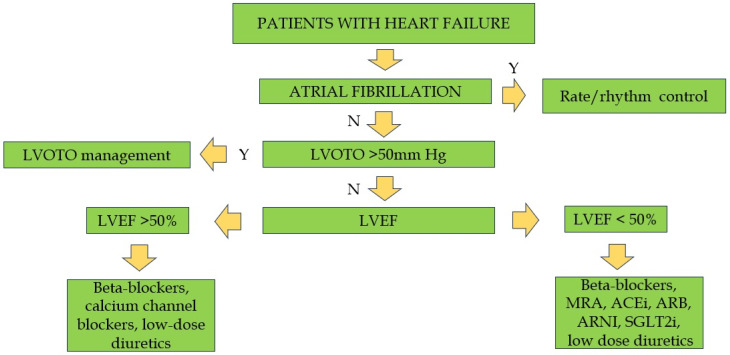
Management of patients with non-obstructive hypertrophic cardiomyopathy. LVOTO = left ventricular outflow tract obstruction; LVEF = left ventricular ejection fraction; ACEi = angiotensin converting enzyme inhibitor; ARB = angiotensin receptor blocker; ARNI = angiotensin receptor neprilysin inhibitor; MRA = mineralocorticoid receptor antagonist; SGLT2i = Sodium-glucose transporter 2 inhibitor; Y = Yes; N = No.

**Table 1 biomedicines-13-01256-t001:** Clinical properties of the main drugs used in the management of obstructive hypertrophic cardiomyopathy. LVOTO: Left ventricular outflow tract obstruction; HF: Heart failure; NSVT: Non-sustained supraventricular tachycardia; SCD: sudden cardiac death, OHCM: Obstructive hypertrophic cardiomyopathy.

Drug	Indication	Starting Dose	Maximum Dose	Notes	Side Effects
**Propranolol**	Angina and dyspnea in patients with or without LVOTO; rate control, ectopic beats	40 mg bid	80 mg bid	Short half time; drug of choice in children	Asthma, bradycardia
**Metoprolol [20]**	Same as propranolol	50 mg qd	100 mg bid	Short half time, Not useful in OHCM	bradycardia
**Bisoprolol [21]**	Systolic dysfunction, HF	1.25 mg qd	15 mg qd	Not useful in OHCO	Asthma, bradycardia
**Atenolol**	Same as propranolol	25 mg qd	150 mg qd	Drug of choice in HCM and hypertension	Hypotension, bradycardia
**Nadolol**	Same as propanolol, reduction in NSVT and SCD when associated with amiodarone	40 mg qd	80 mg bid	Reduction of obstruction	Bradycardia, asthma
**Verapamil [22,25]**	Control of ventricular rate, improvement of diastolic filling	40 mg bid	240 mg bid		AV conduction decrease, peripheral edema
**Diltiazem [22]**	Same as Verapamil	60 mg bid	180 mg bid		Same as Verapamil
**Felodipine**	Refractory angina in HCM	5 mg qd		Useful in microvascular disease	Same as Verapamil
**Disopyramid [27,28,29,30]**	Reduce LVOTO at rest	125 mg bid	250 mg time		QTc prolongation, Anticholinergic effects

**Table 2 biomedicines-13-01256-t002:** Main drugs used in the management of atrial fibrillation in hypertrophic cardiomyopathy.

Drug	Indication	Starting Dose	Maximum Dose	Side Effects
**Disopyramid [117]**	Reduce LVOTO at rest	125 mg bid	250 mg time	QTc prolongation, Anticholinergic effects
**Amiodarone [115]**	AF prevention, control of recurrence SVT/VT, ectopic beats	200 mg qd	200 mg bid	QTc prolongation, thyroid disease, Pulmonary interstitial disease
**Sotalol [116]**	AF prevention, reduction of ectopic beats	40 mg bid	80 mg time	

## Data Availability

Not applicable.

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
