# Peer review of "Integrative Approaches in the Management of Hypertrophic Cardiomyopathy: A Comprehensive Review of Current Therapeutic Modalities"

_biomedicines, 2025, doi:10.3390/biomedicines13051256_

Round 1

Reviewer 1 Report

Comments and Suggestions for Authors

There are various treatment options for HCM, which have been described in detail below. Moreover, evaluation of genetic phenotypes in heart failure is a field that has been attracting attention in recent years. In this respect, too, this manuscript provides useful information to the reader. If you could provide a flow chart or something that would explain which treatment to choose in which patients, it would provide more useful information to readers. Please consider it.

Reviewer 2 Report

Comments and Suggestions for Authors

The authors of the current review paper present the management strategies among patients with obstructive or non-obstructive hypertrophic cardiomyopathy. The paper mainly focused on medical therapies and invasive/surgical approaches. However, there was no statement about the sudden cardiac death risk and ICD implantation as the other face of the HCM management. Please also briefly mention the distinction between HCM and other causes of LV hypertrophy, e.g., amyloidosis, Fabry, etc. Please also look at the other septal ablation techniques like J Invasive Cardiol. 2025 Apr 1. doi: 10.25270/jic/25.00035 and Am J Cardiol. 2023 Mar 1;190:1-7. doi: 10.1016/j.amjcard.2022.11.022. Please also include the role of catheter ablation in patients with HCM and AF besides AADs.       

Reviewer 3 Report

Comments and Suggestions for Authors

In this review article, an attempt is made to review the multimodal treatment approach in hypertrophic cardiomyopathy. The article is well written and supported with data. However, some changes suggested can make it more attractive to readers.

The title needs to be revised

Keywords should be more relevant

In the intro, provide an overview of all the therapies for treating hypertrophic cardiomyopathy.

Any clinical trial in this field must be report

Add limitations and future perspectives

Add references in tables 1 and 2 to make the data more reliable.

Revise Figure 4 with the same dimensions

In Figure 5, Y and N represent what?

Conclusion should be more targetted
